# Modeling Climate Change Effects on Genetic Diversity of an Endangered Horse Breed Using Canonical Correlations

**DOI:** 10.3390/ani14050659

**Published:** 2024-02-20

**Authors:** Carmen Marín Navas, Juan Vicente Delgado Bermejo, Amy Katherine McLean, José Manuel León Jurado, María Esperanza Camacho Vallejo, Francisco Javier Navas González

**Affiliations:** 1Department of Genetics, Faculty of Veterinary Sciences, University of Córdoba, 14071 Córdoba, Spain; carmen95_mn@hotmail.com (C.M.N.); juanviagr218@gmail.com (J.V.D.B.); 2Department of Animal Science, University of California Davis, Davis, CA 95617, USA; acmclean@ucdavis.edu; 3Centro Agropecuario Provincial de Córdoba, Diputación Provincial de Córdoba, 14071 Córdoba, Spain; jomalejur@yahoo.es; 4Instituto de Investigación y Formación Agraria y Pesquera (IFAPA), Alameda del Obispo, 14004 Córdoba, Spain; mariae.camacho@juntadeandalucia.es

**Keywords:** barometric pressure, climate change, endangered autochthonous breed, genetic diversity, gust speed, Hispano-Arabian horse breed, livestock, temperatures, average wind speed

## Abstract

**Simple Summary:**

Gene pools are the foundation for autochthonous breeds’ improved resilience and adaptability to climate-change-derived extreme climatic conditions. Modeling climate change’s impact on the genetic diversity of domestic animals may help predict challenging situations. However, short life cycles and a lack of historical data for extended periods greatly compromise the evaluation of climate change effects. Preserving these domestic resources offers breeding alternatives for those seeking enhanced adaptability. From 1950 to 2019, the long-existing information and international character of the Hispano-Arabian horse breed made it an excellent model for studying other international populations. Wind speed, gust speed, or barometric pressure have greater impacts than extreme temperatures on genetic diversity. Extreme climate conditions may prompt breeders and owners to implement effective short- to medium-term strategies that enhance breed sustainability and the capacity to respond to extreme climate events in the long run. Domestic population preservation can catalyze regulatory changes occurring during breeds’ climate change adaptive process and identify genes that confer greater adaptability while maintaining enhanced performance. This model aids in determining how owners should plan their breeding strategies to obtain more resilient animals adapted to climate-extreme conditions as an efficient alternative to increase profitability and ensure sustainability in these populations.

**Abstract:**

The historical increase in the occurrence of extreme weather events in Spain during the last thirty years makes it a perfect location for the evaluation of climate change. Modeling the effects of climate change on domestic animals’ genetic diversity may help to anticipate challenging situations. However, animal populations’ short life cycle and patent lack of historical information during extended periods of time drastically compromise the evaluation of climate change effects. Locally adapted breeds’ gene pool is the base for their improved resilience and plasticity in response to climate change’s extreme climatic conditions. The preservation of these domestic resources offers selection alternatives to breeders who seek such improved adaptability. The Spanish endangered autochthonous Hispano-Arabian horse breed is perfectly adapted to the conditions of the territory where it was created, developed, and widespread worldwide. The possibility to trace genetic diversity in the Hispano-Arabian breed back around seven decades and its global ubiquity make this breed an idoneous reference subject to act as a model for other international populations. Climate change’s shaping effects on the genetic diversity of the Hispano-Arabian horse breed’s historical population were monitored from 1950 to 2019 and evaluated. Wind speed, gust speed, or barometric pressure have greater repercussions than extreme temperatures on genetic diversity. Extreme climate conditions, rather than average modifications of climate, may push breeders/owners to implement effective strategies in the short to medium term, but the effect will be plausible in the long term due to breed sustainability and enhanced capacity of response to extreme climate events. When extreme climatic conditions occur, breeders opt for mating highly diverse unrelated individuals, avoiding the production of a large number of offspring. People in charge of domestic population conservation act as catalyzers of the regulatory changes occurring during breeds’ climate change adaptive process and may identify genes conferring their animals with greater adaptability but still maintaining enhanced performance. This model assists in determining how owners of endangered domestic populations should plan their breeding strategies, seeking the obtention of animals more resilient and adapted to climate-extreme conditions. This efficient alternative is focused on the obtention of increased profitability from this population and in turn ensuring their sustainability.

## 1. Introduction

Animal genetic diversity is vital for rural livelihoods, enabling breeders to enhance livestock adaptability within existing breeds or create new ones to meet evolving conditions. This context, influenced by societal implications of animal genetic resources, responds to unpredictable abiotic factors like climate change and disease threats [1]. Anthropogenic climate effects have raised the world’s average temperature by 0.6 °C over the past century. Projections suggest this trend will continue, with temperatures expected to rise between 2 °C and 4 °C by mid-century [2]. Such an increase has been mainly ascribed to greenhouse gas emissions [3,4].

Contextually, although wild or feral species may change their distribution range to track new, suitable areas as a response to climate change [5], domestic species are not able to seek their optimal climatic niche as they cannot disperse due to anthropologic barriers [6]. Consequently, it is the responsibility of breeders to adapt to changing conditions, which is crucial for breeds to thrive.

In this sense, the increase in temperatures has been regarded as one of the most significant factors to challenge the adaptability of livestock genetic resources to climate change. However, certain studies have reported breeders may identify other weather parameters, such as wind and rainfall excess, to have even greater repercussions [7].

Climatic events have been reported to impact the genetic diversity of animal species, increasing genetic stochasticity in small populations and, therefore, fixation of poorly adapted haplotypes [8]. Domestic populations act as such small populations in which the limitations are conditioned by anthropological factors. Genetic diversity reflects a breed’s ability to adapt to environmental shifts. While high diversity supports resilience to climate changes, low diversity can limit evolution, leading to decreased productivity and viability. Identifying local populations or breeds more susceptible to climate change is crucial not only for conservation genetics but also for animal welfare and production [9].

According to Reference [9], locally adapted native breeds exhibit a wide range of resilience and plasticity due to their genetic diversity, offering producers valuable options for climate change adaptation. These breeds not only tolerate extreme climatic events but also adapt to resulting challenges, such as food scarcity and emerging diseases. Preserving the gene pool of these breeds is crucial for selecting more adaptable options. Factors like wind speed and barometric pressure have a greater impact on genetic diversity than temperature extremes. Despite challenges in accessing detailed breed-specific data, it is essential for developing accurate climate change models. Long-term breed sustainability and responsiveness to extreme weather events remain paramount considerations [10].

As a result, studies, when performed, can only reliably infer the effect of weather punctual factors rather than permit the actual evaluation of climate change. Furthermore, according to Hoffmann [9], breeds’ environmental envelopes overlap, and their distribution is influenced by production systems, making breed-level predictions or biogeographic models for climate change implications challenging. Breeders address these challenges by selecting native animal genetics or adjusting production environments to suit their domestic populations. These strategies, essential for tackling complex challenges like climate change, raise questions about how animal genetic resources can sustain rural livelihoods while considering broader societal implications.

Contrasting this valuable role of native breeds, Jarvis et al. [11] reported the number of native-breed animals describing a declining worldwide trend. Hence, important efforts must be made to preserve these breeds and their genetic pools, which are the basis for enhanced adaptability potential [12,13,14,15,16]. In this context, the Hispano-Arabian (Há) horse breed is an endangered equine autochthonous breed that was officially recognized within the category of special protection through its inclusion in the Official Catalogue of Spanish Cattle Breeds (Orden APA/2129/2008) [17].

The Há horse breed exemplifies breeders’ strategies to enhance adaptability without sacrificing productivity. Combining the physical endurance and athletic qualities of the Purebred Arabian (PRá) horse with the versatility, predisposition to work, and movement precision of the Purebred Spanish (PRE) horse, the Há breed has effectively mitigated population declines during economic crises in the last century better than its ancestor breeds [18], which granted Há sustainability and extensive distribution across the national territories from north to south [19].

The Há horse breed’s resilience, resistance, and adaptability to adverse conditions stem from its high heterozygosity, enabling its extensive geographical spread. This breed thrives in a diverse range of Spanish countryside ecosystems, from below-zero Cantabrian mountains to temperatures exceeding 40 °C in regions like the Guadalquivir River Valley, Dehesa forests, and marsh areas [18]. As a result, the Há horse breed plays a pivotal role in the productive and environmental conservation of the aforementioned ecosystems as a valuable element in the labor of managing livestock, mainly bovine animals, while acting as a complementary source of income in agricultural and livestock farms [20,21].

Current species vulnerability and biogeographical shift models frequently neglect adaptive genetic variation. However, the Spanish Union of Hispano-Arabian Horse Breeders has facilitated the integration of genealogical registries into models. This integration enables the assessment of climate change impacts on breed diversity and the prediction of future changes. Recognizing phenotypic plasticity and genetic adaptations in native breeds is crucial for their profitable sustainability and persistence. As suggested by References [22,23], incorporating intraspecific domestic adaptations in vulnerability assessments to climate change becomes especially relevant to plan conservation management strategies [24]. However, there are still no methodologies that relate breed information with the existing and changing climatic conditions over considerably long periods of time [25], which in turn translates into the existing scarcity of observational and experimental studies of the local climate [26].

This study aims to assess the impact of climate change on the genetic diversity of an endangered nationally adapted horse breed in various Spanish climatic conditions. Regularized canonical correlation analysis (RCCA) was employed to analyze the relationship between genetic diversity and climate events from 1950 to 2019. This analysis informs the development of predictive models for managing genetic diversity in response to future climate changes. This methodology can serve as a model for studying climate change effects on domestic animal populations internationally.

## 2. Materials and Methods

### 2.1. Data Registries, Pruning, and Software Tool

Since the Hispano-Arabian (Há) horse breed is the product that results from the cross between Spanish (PRE) and Arabian (PRá) purebred horses, the historical pedigree files for the three breeds were considered to build the historical pedigree database used in this study. The historical dataset comprised a total of 207,100 horses.

The Há historical pedigree database file was supplied by the Spanish Union of Purebred Hispano-Arabian Horses Breeders (UEGHá) and comprised 11,010 individuals, 4268 males and 6742 females, born between January 1950 and April 2019. The PRE historical pedigree database was supplied by the National Association of Purebred Spanish Horse Breeders (ANCCE) and comprised 172,797 individuals—83,408 males and 89,389 females, born between January 1884 and July 2019. The Spanish PRá historical pedigree record was provided by the Spanish Association of Arabian Horse Breeders (AECCA). The PRá historical pedigree database comprised a total of 23,293 individuals—11,143 males and 12,150 females, born between January 1898 and June 2019. Genetic diversity parameters were calculated in a previous study by Marín et al. [27]. Only those genetic diversity parameters that could be calculated per individual were considered: genetic diversity parameter (GCI), coancestry (C, %), non-random mating degree (α), number of maximum, complete and equivalent generations, and individual inbreeding ΔF.

The PRE and PRá pedigree databases were excluded from this study due to the impossibility of accessing the information relative to the animal’s exact birth location, which thus hindered the evaluation of the repercussions of climate change on genetic diversity parameters. As a result, all climate change association analyses were performed on a population set only comprising the Há animals in the historical pedigree database file for which birth location could be determined precisely. A pruning process was carried out to define the study sample discarded, after which 187 individuals were discarded from the initial 11,010 Há horses that had been used to calculate genetic diversity parameters. A number of 89 individuals were discarded because their birthplace was not registered appropriately; 92 were born abroad, hence their exact birth location had not been registered but their country of origin (Sweden, Germany, Bulgaria, Belgium, France, UK Denmark); 2 were discarded due to lack of climatic registries in their province of birth (1900, Cádiz); and 4 were discarded to province birth location inaccuracies. The final study database comprised 10,823 individuals—4188 males and 6635 females, born between January 1950 and May 2019.

### 2.2. Inbreeding, Coancestry, Non-Random Mating Degree, and Genetic Conservation Index

Individual inbreeding (F) was computed using the methods in Meuwissen and Luo [28]. Each individual’s average relatedness (AR) was calculated according to Gutiérrez, Marmi, Goyache, and Jordana [12]. According to Leroy et al. [29], F and coancestry (C) coefficients are identity estimators by descent (IBD), a probability that differs whether the alleles considered belong to a single individual or two individuals. The individual rate of inbreeding ($\bar{∆F}$) for the generation is calculated according to Gutiérrez et al. [30] through ${∆F}_{b}=1-\sqrt[{t_{b}-1}]{1-F_{b}}$, where t_b_ is the number of complete equivalent generations, and F_b_ is the inbreeding coefficient of the individual b. The individual rate of coancestry ($\bar{∆C}$) for the generation was computed following the methods of Cervantes et al. [31] through $C_{ba}=1-\sqrt[{\frac{tb+ta}{2}}]{1-C_{ba}}$, where t_b_ and t_a_ are the numbers of equivalent complete generations, and C_ba_ is the coancestry coefficient for the individuals b and a. The degree of assortative mating (non-random mating of individuals having more genetic or phenotypic traits in common than likely in random or disassortative mating) was computed following the methods of Caballero and Toro [32] through $\left(1-F\right)=(1-C)(1-\alpha )$ [33]. The GCI (Genetic Conservation Index), which is the effective number of founding ancestors that integrate each pedigree, was calculated by following the methodology used by Oliveira et al. [34] and Marín Navas et al. [35], according to the formula GCI = 1/Σ P_i_^2^ y P_i_ = Σ (1/2)^n^, where P_i_ is the proportion of genes of the founder i in the pedigree of a certain animal, and n the number of pathways from the founder to the animal object of the study. All genetic analyses were performed using ENDOG (v4.8) software [12] on the historical pedigree (both living and dead individuals).

### 2.3. Pedigree Completeness Index

We studied the number of births to compute the total number of offspring per stallion or mare. The pedigree completeness index (PCI) was computed using the maximum number of generations traced (number of generations separating an individual from its earliest ancestor); the number of complete traced generations (both known ancestors furthest generation); the average number of complete equivalent generations (all known ancestors addition, calculated as (1/2n), where n is the number of generations setting the individual apart from each known ancestor [36], equal to $\sum_{a=1}^{n_{b}}\frac{1}{2^{g_{ab}}}$, where n_b_ is the total number of ancestors of the animal, and b and g_ab_ are the number of generations between b and its ancestor a) [37].

### 2.4. Meteorological and Moon Cycle Records

Historical day records for altitude, average, minimum and maximum temperature, rainfall per day, wind direction, wind speed, gust speed, sunlight hours, and maximum and minimum barometric pressure were obtained from the State Meteorological Agency (AEMET) (http://www.aemet.es/) (accessed on 16 February 2023).

### 2.5. A Priori Assumptions

Regularized canonical correlation analysis (RCCA) assumes linearity and thus a linear relationship between the canonical variates and each set of variables. However, RCCA only reports a valid inference when appropriately applied if three assumptions are met [38].

First, as in other multivariate test statistics, RCCA requires that the variables have a multivariate normal distribution in the population [38]. As supported by Hair et al. [39] and Razali [40], the Shapiro–Wilk test was used as it is the most powerful test for all types of distribution and sample sizes (*n* > 5000). The normality test function of the Describing Data Menu of XLSTAT 2014 (Pearson Edition) (Addinsoft, Paris, France) was used to evaluate normality. The Shapiro–Wilk test revealed the variables in both sets (genetic diversity parameters and climate-related variables) not to be normally distributed. Considering these results, the D’Agostino–Pearson test was performed to determine whether data had been sampled to a normally distributed population. The D’Agostino–Pearson test was performed using Excel version 2016 and the DAGOSTINO and DPTEST from the Real Statistics Resource Pack. The D’Agostino–Pearson’s test reported data were sampled from a normally distributed population (*p* > 0.05).

The homoscedasticity assumption must also be checked [41]. For this, Levene’s test was used with the k-sample comparison of variances function of the Parametric Tests menu of XLSTAT 2014. As homoscedasticity was not met (*p* < 0.05), we decided to use permutation tests’ inference for canonical correlations, as suggested by Winkler [42].

Permutation tests’ inference for canonical correlations has been reported to address the aforementioned difficulties with well-known advantages. For instance, no underlying distributions need be assumed, non-independence and even heteroscedastic variances can be accommodated, non-random samples can be used, and a wide variety of test statistics are allowed.

In this context, as with all least-squares procedures, outliers must be evaluated. The identify outliers procedure of the Analyze/Built-In analysis of the Column Analyses package of GraphPad Prism version 9.0 was used to detect the likelihood of outliers that could distort fitting properties. To this aim, the ROUT method was implemented as it combines robust regression and outlier removal and can be used to fit a curve that is not influenced by outliers. The ROUT method is based on the false discovery rate (FDR). A Q-coefficient level, which is the maximum desired FDR, must be predefined to determine how aggressively the ROUT method defines outliers [43]. Unless you have a strong reason to choose otherwise, the default value of 1% is recommended. If there really are outliers present in the data, Prism will detect them with a false discovery rate of less than 1%. A Q coefficient of 1% was used, as suggested by Martins-Bessa et al. [44]. No outliers were detected for any parameter studied.

Second, similar to multivariate regression, canonical correlation analysis requires a large sample size [41]. In this regard, some authors explain a minimum of ten cases per variable is needed, although such a requirement may diminish as the sample grows [45].

Third, the occurrence of a curvilinear relationship will reduce the effectiveness of the analysis. In this particular case, the multicollinearity assumption was tested to ensure redundancies in the variables considered do not affect the structure of the matrices or overinflate variance explanatory potential. The variance inflation factor (VIF) was computed and used as an indicator of multicollinearity. A recommended maximum VIF value of 5 was suggested by Marin et al. [18]. The VIF was computed using the Multicollinearity Statistics function of the Describing Data menu of XLSTAT 2014.

### 2.6. Regularized Generalized Canonical Correlation Analysis (RCCA)

Regularized canonical correlation analysis (RCCA) was performed using the Canonical Correlation Analysis function of the Multiblock Data Analysis menu of XLSTAT 2014 [46] and the SPSS MANOVA syntax described in Appendix A S1 in SPSS version 25.0 [47].

### 2.7. Pearson’s Product–Moment Correlations

The first step after multicollinearity evaluation was to determine the correlation within the sets of genetic diversity parameters and climate-related traits and between each other. The Pearson product–moment correlation coefficient was chosen as a part of the routines in RCCA when run in XLSTAT 2014. The Canonical Correlation Analysis function of the Multiblock Data Analysis menu of XLSTAT 2014 was used to test Pearson’s product–moment correlation. Guidelines to interpret Pearson’s correlation coefficients can be found in the work of Profillidis and Botzoris [48], as suggested by Gil Lebrero et al. [49].

### 2.8. Validity

Regularization of CCA has been suggested for cases in which there are sample size limitations and high data dimensionality to prevent inaccurate estimates and data overfitting issues [50]. Regularized CCA has been used [51] and results robust when disclosing linear relationships between two sets of variables when the aforementioned incidences are not present. As suggested by Olson [52], Pillai’s trace is recommended for general use as one or more of these assumptions being violated (residuals normality, equal variance–covariance matrices of each group of residuals, independence of observations) tends to be the most robust test statistic. Tests of significance of all canonical correlations (Pillais’ trace criterion) were calculated using the Canonical Correlation function of the Multivariate Analysis menu and the MANOVA multivariate regression and related procedure set in STATA version 16 [53].

### 2.9. Variability Explanation

Afterwards, eigenvalues were calculated. The eigenvalues derive from the product of the model matrix and the inverse of the error matrix and can be calculated using the squared canonical correlations. The largest eigenvalue is equal to largest squared correlation/(1 − largest squared correlation). The relative size of the eigenvalues reflects how much of the variance in the canonical variates can be explained by the corresponding canonical correlation. Thus, the eigenvalue corresponding to the first correlation is greatest, and all subsequent eigenvalues are smaller.

### 2.10. Canonical Correlations

Canonical correlations can be interpreted as any other Pearson correlations (see rule of thumb interpretation criteria in the work of Profillidis and Botzoris [48], as suggested by Gil Lebrero et al. [49]. Canonical correlations range between –1 and 1: a value near 0 indicates low correlation, and an absolute value near 1 indicates near-perfect correlation [41]. Meaningful variates are detected when their canonical correlation is ≥0.30, which corresponds to about 10% of the explained variance. All meaningful and interpretable canonical correlations should be reported, despite reporting only the first dimension being common in research, as suggested by Marin et al. [18].

The redundancy coefficients show that a small proportion of the variability of the input variables is predicted by the canonical variables. That is, the square of the correlation represents the proportion of the variance in one group’s variance explained by the other group’s variance.

### 2.11. Roots

Roots are the rank of the given eigenvalue (largest to smallest). There are as many roots as there were variables in the smaller of the two variable sets (Genetic diversity parameters set). This is the set of roots included in the null hypothesis that all of the correlations associated with the roots in the given set are equal to 0 in the population. By testing these different sets of roots, we can determine how many dimensions are required to describe the relationship between the two groups of variables. Because each root is less informative than the one before it, unnecessary dimensions will be associated with the smallest eigenvalues.

### 2.12. Wilks’ Lambda and R^2^

Each Wilks’ lambda value can be calculated as the product of the values of (1 − canonical correlation) for the set of canonical correlations being tested. The Wilks’ lambda value for the canonical correlation of this report is the multivariate generalization of R^2^. The Wilks’ lambda statistic is interpreted just the opposite of R^2^. A value near 0 indicates high correlation, while a value near 1 indicates low correlation [41]. Smaller values of Wilks’ lambda indicate the greater discriminatory ability of the function.

### 2.13. Canonical Correlation Analysis k-Fold Cross-Validation

Ten-fold cross-validation of canonical correlation analysis was performed to determine the validity, which reduces the contamination of results due to sample-specific error of the RCCA [54]. The use of k = 10 was chosen as recommended for load estimations since this value was associated with better performances and less biased estimates [55]. R 4.1.1 for Windows, RStudio Version 1.4.1717, was used to perform 10-fold cross-validation using the RStudio software. Ten-fold cross-validation of canonical correlation analysis was performed using CCA: An R Package to Extend Canonical Correlation Analysis by González et al. [56] and the mixOmics 6.16.3 [57] packages for R in R Studio Version 1.4.1717 (http://www.R-project.org) (accessed on 9 January 2024) [58].

Regularization parameters (λ_1_ and λ_2_) were estimated using the tune.rcc function [59], where a set of positive values were chosen to evaluate the cross-validation (CV) score for each point in the network, achieving an optimal value for λ_1_ and λ_2_ that offered the highest CV score.

## 3. Results

### 3.1. A Priori Assumptions

The outputs of multicollinearity analysis revealed the fact that the variables of non-random mating degree, complete and equivalent generations from the genetic diversity parameter set and average temperature and maximum barometric pressure from the climate-related parameters set should be discarded due to their redundancies with other variables (VIF ≥ 5) in the analysis. The coancestry variable was discarded as well due to singularity (Table 1).

### 3.2. Regularized Generalized Canonical Correlation Analysis (RCCA)

#### Pearson’s Product–Moment Correlations

Among the genetic diversity parameters, the offspring variable exhibited a weak negative linear correlation with the other parameters (0.000 < r_XY_ < −0.300), except for the individual increase in inbreeding rate (ΔF, %), which showed a weak positive correlation (0.000 < r_XY_ < 0.300). Inbreeding (F, %) demonstrated a weak to moderate (−0.281< r_XY_ < 0.300) linear correlation with relatedness coefficient (ΔR, %), genetic conservation index (GCI), and maximum number of generations. These correlations were negative and stronger, albeit still moderate, with individual increase in inbreeding rate (ΔF, %) (−0.300 < r_XY_ < −0.600). Conversely, the linear relationship between inbreeding (F, %) and individual increase in inbreeding rate (ΔF, %) was moderate to strong and positive (r_XY_ = 0.774) (Table 2).

Altitude showed a moderate negative correlation with minimum temperature (−0.300 < r_XY_ < −0.600), while weak negative correlations were observed between altitude and rainfall, maximum temperature, and average wind speed (0.000 < r_XY_ < −0.300). Conversely, weak positive correlations were found between altitude and other variables (gust speed, sunlight hours, minimum barometric pressure, wind direction) (0.000< r_XY_ < 0.300). Rainfall exhibited weak positive correlations with minimum temperature, wind direction, average wind speed, gust speed, and minimum barometric pressure (0.000 < r_XY_ < 0.300). However, it showed weak to moderate negative correlations with sunlight hours and maximum temperature (r_XY_ ≥ 0.174). Minimum temperature displayed weak positive correlations with most variables, except for a moderate/large correlation with maximum temperature. Maximum temperature showed weak positive correlations with wind direction, average wind speed, and gust speed, but moderate correlations with sunlight hours and minimum barometric pressure (0.300 < r_XY_ < 0.600). The most notable negative correlation was a moderate one between average wind speed and wind direction (−0.300 < r_XY_ < −0.600). Weak positive correlations were found between average wind speed and gust speed, and minimum barometric pressure (0.000 < r_XY_ < 0.300), while a weak negative correlation was found between average wind speed and sunlight hours (0.00 < r_XY_ < −0.300). Similarly, weak positive correlations were observed between gust speed and sunlight hours, and minimum barometric pressure (0.000 < r_XY_ < 0.300), while a positive moderate correlation was found between sunlight hours and minimum barometric pressure (0.300 < r_XY_ < 0.600) (Table 3).

Relatedness coefficient (ΔR, %), genetic conservation index (GCI), and maximum number of generations showed positive and weak correlations (0.000 < r_XY_ < 0.300) with various climate-related variables. The strongest correlation, with a coefficient of r_XY_ = 0.207, was observed between minimum barometric pressure and maximum number of generations. Conversely, average wind speed exhibited weak and negative correlations with the same climate-related variables (0.00 < r_XY_ < −0.300). Inbreeding (F, %) and individual increase in inbreeding rate (ΔF, %) displayed similar trends, with slightly stronger correlations reported for the latter, especially with all climate-related variables (twice the values for inbreeding). However, the linear relationship between inbreeding (F, %) and individual increase in inbreeding rate (ΔF, %) was weak but positive for average wind speed (0.000 < r_XY_ < 0.300). Offspring showed a negative and weak linear correlation with all climate-related variables, except for minimum barometric pressure, where the correlation was negligible (r_XY_ = 0.002) (Table 4).

### 3.3. Validity and Variability Explanation

Table 5 reports Pillai’s trace criterion to be highly statistically significant (*p* < 0.01); hence, the RCCA is valid. As shown in Figure 1, the first variate (F1) alone explains 76.28% of the variability in both datasets. This figure shows the six eigenvalues for each of the variates determined.

### 3.4. Canonical Correlations and Roots

As shown in Table 6 and Appendix A, the first pair of variates, a linear combination of the genetic diversity parameters and a linear combination of the climate-related traits, has a correlation coefficient of 0.339. The second pair has a correlation coefficient of 0.166, and the third pair 0.071.

Each subsequent pair of canonical variates is less correlated. Figure 2 represents the standardized canonical coefficients; that is, if all of the variables in the analysis are rescaled to have a mean of 0 and a standard deviation of 1, the coefficients generating the canonical variates would indicate how a one-standard-deviation increase in the variable would change the variate without depending on the units in which every specific variable is measured. This is of special relevance when variables measured in different units are clustered together, as it happens in our study. After the evaluation of standardized coefficients, the resulting discriminant functions for the climate-related variable set were as follows:F1:0.531 × Height + 0.088 × Rainfall + 0.238 × Minimum Temperature + 0.248 × Maximum Temperature + 0.509 × Wind Direction + 0.169 × Average Wind Speed + 0.123 × Gust Speed + −0.037 × Sunlight hours + 0.315 × Minimum Barometric PressureF2:−0.515 × Height + 0.186 × Rainfall + −0.502 × Minimum Temperature + 0.583 × Maximum Temperature + 0.085 × Wind Direction + 0.782 × Average Wind Speed + −0.177 × Gust Speed + 0.066 × Sunlight hours + 0.212 × Minimum Barometric Pressure

While for the genetic diversity parameter set, they were as follows:F1:0.097 × Inbreeding (F. %) + −0.217 × Relatedness Coefficient (ΔR. %) + 0.105 × Genetic Conservation Index (GCI) + 0.973 × Maximum number of Generations + −0.232 × Individual Increase in Inbreeding Rate (ΔF. %) + −0.044 × OffspringF2:0.614 × Inbreeding (F. %) + −0.169 × Relatedness Coefficient (ΔR. %) + −0.974 × Genetic Conservation Index (GCI) + 1.038 × Maximum number of Generations + 0.008 × Individual Increase in Inbreeding Rate (ΔF. %) + −0.039 × Offspring

Figure 2 suggests that the variable offspring is correlated and correlated negatively with Variates 1 and 2. They are also anti-correlated with the minimum and maximum temperature, sunlight hours, minimum barometric pressure, and rainfall. This means that with climatic parameters such as minimum and maximum temperature, sunlight hours, minimum barometric pressure, and rainfall, the number of offspring reduces. Oppositely, the variables of inbreeding (f) and individual increase in inbreeding rate (ΔF) and average wind speed are correlated positively with Variates 1 and 2. They are also anti-correlated with the other diversity parameters such as genetic conservation index (GCI), average relatedness (ΔR) and number of maximum generations, and climatic-related variables such as altitude, gust speed, and wind direction.

### 3.5. Wilks’ Lambda and R^2^

All the variates F1, F2, F3, F4, F5, and F6 are significantly linked to initial correlation tables (Table 7). Multivariate generalization of R2 for maximum number of generations, GCI, and ΔR suggests these factors accounted for the greatest intracluster and intercluster explanatory potential of either diversity or climatic data variability (from 5.2 to 11% for ΔR and maximum number of generations, respectively) (Table 8).

### 3.6. Canonical Correlation Analysis k-Fold Cross-Validation

Results for 10-fold cross-validation for RCCA are shown in Figure 3. As we can observe, cross-validation coefficients in Figure 3 match those values for Wilks’ lambda at the for F1 in Table 6. Hence, we can conclude the present analysis is valid, and no distortion derived from sample-specific error can be presumed. The regularization process permits choosing optimal values for λ_1_ and λ_2_ parameters. These parameters were selected by a 10-fold cross-validation procedure on a regular grid of size 100 × 100 defined on the region 0.001 < λ_1_ < 1, and 0.001 < λ_2_ < 1 and resulting in λ_1_ = 0.7502 and λ_2_ = 0.001 (coordinates of the white section depicted in Figure 3) following the prescriptions in [60]. The optimal cross-validation score was 0.341. The choice of dimensions d (1 ≤ d ≤ *p*) to include in the further analysis was performed according to [51], who suggests an empirical approach based on the inspection of the plot of canonical correlations versus dimensions and on the selection of the appropriate number of dimensions before a clear gap among canonical correlations. On the basis of the obtained λ_1_ and λ_2_ parameters, a clear gap was observed between the first and the second canonical correlations (Table 6 and Appendix A), which agrees with the fact that 92.85% of variance was captured by the first two dimensions (F1: 76.28% and F2: 16.57%; Figure 2).

## 4. Discussion

### 4.1. Genetic Diversity Parameters’ Inclusion

The values obtained for VIF ≥ 5, thus the decision to include or exclude genetic diversity parameters, was supported by certain principles. The coefficient of relatedness, being twice the kinship or coancestry coefficient, likely explains the observed singularity in coancestry [61], leading to its exclusion. The relationship between inbreeding and coancestry supports the redundancy of the non-random mating degree, as these variables are necessary for its calculation using the formula nrm = (1 − F)/(1 − C) [62].

The information provided by complete and equivalent generations overlaps with the maximum number of traced generations, representing the number of generations between an individual and its farthest known ancestor, even if that generation is incomplete due to unknown ancestors [63].

Some authors [64] have suggested that the maximum number of traceable generations may not reliably indicate gaps in the pedigree while using equivalent generations [37] or the number of generations in a comparable complete pedigree does. This highlights the importance of considering potential biases stemming from uneven pedigree knowledge of sire and dam pathways in diversity studies [65]. However, leveraging maximum traceable generations alongside other diversity parameters like inbreeding rates or coancestry rates helps mitigate the limitations caused by incomplete data, enhancing the robustness of diversity analyses.

Indeed, according to González-Recio et al. [66], complete equivalent generations may not accurately reflect inbreeding effects due to their reliance on incomplete or poor pedigree information, particularly common in endangered breeds. This limitation is circumvented with maximum generations, which adjust inbreeding rates based on pedigree depth, serving as an indicator of inbreeding increment for each animal regardless of the number of known generations [66]. 

### 4.2. Climatic Conditions Parameters’ Inclusion

Average temperature redundancies may stem from the inclusion of minimum and maximum temperatures in the model. Some authors [67] propose that including average temperatures could be more insightful, considering that maximum and minimum temperatures are often separated by several hours, potentially affecting the reliability and repeatability of results. However, limitations of these models may arise from assuming the daily temperature curve to be symmetrical about 12 h [68]. Thus, including the time between maximum and minimum temperatures may better represent day-length changes throughout the year. Additionally, Droogers and Allen [69] suggest that in situations with expected low accuracy in weather measurements, opting for a limited data set or focusing on collecting only maximum and minimum temperature and precipitation data may be preferable over attempting to establish a full meteorological station data evaluation due to the greater reliability of these parameters. Regarding maximum barometric pressure redundancies, previous literature has used wind power and maximum barometric pressure to measure storm intensity [70,71,72,73]. However, according to Mendelsohn et al. [73], suggest that minimum barometric pressure may be a more accurate predictor of storm damages than maximum wind speed, indicating its better performance in explaining climate data variability. The coefficient of wind speed and minimum pressure suggests that storm damage is a highly nonlinear function of storm intensity, with increases in wind speed and decreases in minimum pressure leading to doubled damages.

This idea is supported by other authors [72], who suggest that large-scale atmospheric and ocean processes may be driven by an ‘inverted barometer effect’, considering only annual mean and maximum values of barometric pressure. This “inverted barometer effect” may characterize warming events and be preceded by declining barometric pressures, weakening winds, and moderate increases in relative humidity. Thus, the greater relevance of minimum barometric pressure oscillations compared to maximum barometric pressures may indicate the latter’s confounding effect due to its intrinsic correlation with the more representative minimum barometric pressure [74] and the need to discard it from the model.

### 4.3. Genetic Diversity Parameters’ Intracluster Correlations

Multivariate analysis revealed that maximum number of generations, GCI, and ΔR were the most significant factors, explaining 5.2% to 11.1% of the variability in diversity and climatic data (Table 3 and Table 7). Strong positive correlations among these diversity factors (Pearson’s ρ ≥ 0.724) further supported their importance (Table 2). Following closely were F and ΔF, contributing 2.7% to 4.8% to the explanatory potential (Table 8). This aligns with the strong positive correlation (Pearson’s ρ = 0.774) found between F and ΔF (Table 2).

The correlations between the maximum number of generations, GCI, and ΔR with F and ΔF were moderately strong and negative (Table 2), ranging from −0.506 to −0.406 for ΔF and −0.308 to −0.281 for F. This negative relationship may stem from the differential effects of pedigree completeness on the accuracy of determining genetic diversity parameters. Studies in horses [75,76] have found that despite efforts to avoid close matings, nearly all horses with complete pedigrees are related within the first five ancestral generations, leading to some degree of inbreeding.

Generally, individuals belonging to endangered horse breeds are complexly interrelated through multiple remote ancestors, often reaching the founders of the population (for whom genealogy is at best partially registered). Diversity parameters such as F, GCI, and ΔR remarkably increase as pedigree depth does. In this context, diversity parameters intracluster correlation may be ascribed to the fact that, as suggested by Iglesias Pastrana et al. [77], the estimates derived from the analyses of non-robust pedigrees (i.e., low depth, missing information, errors, unknown founder relationships, among others) may be favored if empirical estimates of relatedness (C and/or ΔR, for instance) are determined. These authors continued explaining that this favorable effect may be restricted to small-sized threatened populations, with limited or missing genealogical background, in which the proportion of polymorphic loci is commonly small, which may compel the use of large numbers of genomic markers when seeking such an enhanced estimation accuracy, which is not our case.

When large populations are evaluated, these limitations may translate to allele frequencies of the historical population being unknown, which may bias the inference of F as a direct consequence of potential changes occurring due to genetic drift. As a result, identity by descent (IBD) and identity by state (IBS) probabilities underlying genetically mediated similarities among relatives may not be distinguished, which is a drawback that can only be saved if the information comprised in pedigrees increases in quality (completeness).

Despite the theoretical expectation that inbreeding (F) and coancestry would asymptotically converge over generations, findings from a study on Standardbred horses by MacCluer et al. [76] suggest otherwise. They found that while inbreeding may increase noticeably in the first few generations of current animals, it levels off after around 10–12 generations, even with enhanced pedigree registration practices. However, this leveling off does not occur for other parameters like genetic conservation index (GCI), individual increase in relatedness (ΔR), or coancestry (C) in real populations [77,78,79]. This lack of asymptotic behavior in certain parameters can be attributed to population structuring caused by factors such as geographical distance, selection, or local founder effects. Population structure remains permanent yet incomplete, with selection intensity varying over time depending on the success of newly selected families, reproductive enhancements, or changes in herds’ structure. This structuring process is often cryptic, making it challenging to identify individuals sharing a significant proportion of genes from selected families, especially in populations where only a few large herd owners contrast with a vast number of owners who own just one or two individuals.

Caballero [80] suggests that in structured populations, estimating population size based on coancestry may be more accurate than using inbreeding, as coancestry is less affected by population structure [81]. Population structuring, often due to selection practices, can lead to a rapid reduction in genetic diversity and variability, known as Bulmer’s disequilibrium effect. Unlike reductions caused by drift or inbreeding, this effect is reversible once the selection regime ends [82]. Additionally, positive assortative mating can further reduce variability by creating a non-zero covariance between parental breeding values [83]. In real horse populations, minimizing coancestry between mating individuals helps maintain allelic frequencies and prevents losses of gene diversity [84].

In horse populations, estimating effective population size based on coancestries can provide valuable insights into the population’s genetic diversity and structure [31]. While recent studies have presented contradictory findings regarding the accuracy and sensitivity of effective size estimates derived from coancestry increases [79], there is a consensus on the importance of optimizing parental contributions to manage genetic diversity. This involves minimizing global coancestry weighted by individual contributions, particularly when considering other diversity parameters like individual increase in inbreeding, which can still be influenced by population structure [85,86]. Inaccurate estimation of diversity parameters may reduce the effectiveness of diversity protection measures [79]. The lack of correlation between offspring number and other diversity parameters suggests that individual prolificacy is not necessarily linked to mating preferences for less or more related animals. This is supported by low average levels of non-random mating degree, inbreeding, or coancestry in the population.

### 4.4. Climatic Conditions Parameters’ Intracluster Correlations

Spain has a history of increased occurrence of extreme weather events across various climatic conditions, particularly notable in the past thirty years [87,88,89]. These events not only shape mean climate conditions but also alter the probability of extreme events, making Spain an ideal location for studying climate change effects due to its climatic diversity and the varied impact of extreme conditions over time [90].

Analysis of correlations in this study indicates strong and variable interrelationships between climate conditions and extreme events. For instance, the probability of hot extremes doubles with a 2 °C global warming compared to 1.5 °C, while temperature fluctuations due to climate change nonlinearly affect rainfall extremes, resulting in a 20% increase in precipitation per each Celsius degree rise [91]. Moreover, climate change can influence other climatic factors like wind speed and sunlight hours, with wind speed showing an increasing trend and wave height, although less pronounced, also exhibiting changes, particularly in extreme events [92].

### 4.5. Genetic Diversity and Climatic Conditions Parameters’ Canonical Correlations

When evaluating domestic animal populations, their response to climate change events depends not only on their inherent adaptation capabilities but also on the strategies employed by owners to mitigate potential adverse impacts [93]. For instance, climate-induced changes in crop cycles may compel owners to adjust by either temporarily or permanently relocating their herds in search of more suitable feeding resources [94]. Such biogeographical migrations are a global response to climate change driven by the necessity to sustain, afford, and profit from domestic populations. Climate factors like prolonged sunlight exposure, increased temperatures, or altered wave patterns affect crop rhythms, influencing their availability as animal feed and thus shaping husbandry practices [95]. Consequently, domestic animals may need to adapt to both climatic shifts and changes in husbandry practices initiated by owners, which could entail adjustments in breeding plans, reproductive strategies, or geographical movements [96,97,98].

Changes in usual husbandry practices often lead to population bottlenecks. In this study, the altitude at which animals were bred was monitored, as it is a known influencer of weather conditions. Its inclusion in the model aimed to assess whether vertical movements of owners from sea level occurred over the decade and if these movements affected diversity in the Há domestic population [99]. The results indicate that as owners relocate their animals to higher altitudes (averaging around 400 m above sea level over the last 40 years) (Figure 4), the likelihood of isolation or poor connectivity increases, particularly in atomized populations such as certain endangered horse breeds, where the ratio of owners to population members is nearly one-to-one.

The isolation process in domestic populations is driven by bottlenecks resulting from owners’ artificial choices aimed at optimizing their activities at lower costs, often favoring more profitable individuals. This selective pressure is particularly pronounced in endangered populations managed by small to medium holders, who may prioritize immediate economic gains over long-term genetic diversity preservation, exacerbating the loss of valuable genetic diversity.

Our study revealed strong correlations between genetic diversity parameters and environmental factors such as minimum barometric pressure, wind speed, Gust Speed, and altitude, with weaker correlations observed for maximum and minimum temperature and sunlight hours (Figure 4, Figure 5 and Figure 6). Gusts are rapid fluctuations in wind speed that typically occur during storms or strong weather systems. The relationship between gusts, wind direction, and rainfall is complex and can vary depending on the specific weather conditions and geographical location.

In general, gusts are often associated with strong winds, which can occur in various weather patterns, including thunderstorms, hurricanes, or even localized squalls. These strong winds can result from pressure gradients, temperature differentials, or other atmospheric dynamics.

Wind direction can influence the distribution and intensity of gusts. For example, in the context of a thunderstorm, downdrafts associated with intense rainfall can produce gusty winds that descend rapidly to the surface. Wind direction can also affect how gusts are experienced in different locations; for instance, if a storm is moving from west to east, areas to the east of the storm’s center may experience stronger gusts as the system passes.

Rainfall itself may not directly cause gusts, but it can be associated with the same weather systems that produce gusty winds. In some cases, heavy rainfall can be accompanied by strong winds, leading to turbulent conditions and potentially enhancing gustiness.

Overall, the relationship between gusts, wind direction, and rainfall is intertwined within the broader context of atmospheric dynamics and weather patterns, making it essential to consider multiple factors when analyzing their interactions. The relationship between gusts and wind direction in degrees involves understanding how gusts interact with the prevailing wind direction. Generally, gusts occur as rapid increases in wind speed superimposed on the prevailing wind. Wind direction is typically measured in degrees clockwise from true north, with 0 degrees representing north, 90 degrees representing east, 180 degrees representing south, and 270 degrees representing west.

The relationship between gusts and wind direction may vary depending on the specific weather conditions in the area where the horses live, which means wind currents may likely describe area-specific wind routes, which are described by gusts as well. For instance, in many cases, gusts may occur perpendicular to the prevailing wind direction. For example, if the prevailing wind is blowing from the north (0 degrees), gusts may occur from various directions but tend to be most pronounced from the west (270 degrees) or east (90 degrees).

This suggests that climatic factors other than temperature play a significant role in shaping genetic diversity, possibly because owners are more adept at mitigating the effects of extreme temperature events or because individuals may rather be more genetically equipped to cope with them. Gust speed, average wind speed, and barometric pressure, in particular, have been identified as important indicators of changes in solar power production [100], indicating their potential implications for climate change [101]. Additionally, these factors can influence ecological aspects of animal populations, such as reproductive cycles and behavioral and migratory patterns, in various species [102,103].

The ecological adaptability of domestic populations is inherently constrained by anthropogenic factors such as housing and handling practices, which may influence animals’ responses to challenging situations. Owners’ habits and adaptive strategies can mitigate these challenges to some extent [96]. Strong correlations were observed between genetic diversity parameters such as ΔR, GCI, and the number of maximum generations (ρ ≈ 0.100 to 0.200). These parameters indicate breeders’ increased interest in genealogical information and the utilization of animals with a higher effective number of founder ancestors, contributing to the conservation of the gene pool and the presence of climate change adaptability-enhancing genes. The rise in relatedness coefficients suggests intentional mating practices aimed at maintaining genetic diversity, as evidenced by the negative correlations between the inbreeding coefficient (F) and changes in F (ΔF) (ρ ≈ −0.150 to −0.100). Breeders may prioritize mating between genetically diverse animals to preserve diversity while maintaining the influence of founders in pedigrees. Interestingly, the number of offspring did not correlate with these parameters, indicating that owners and breeders prioritize the quality of animals over quantity, especially in extreme climatic conditions where quality fodder may be scarce or expensive, affecting profitability.

Extreme climate conditions, rather than gradual climate shifts, may prompt breeders to implement short- to medium-term strategies that yield long-term effects. Owners can facilitate climate adaptation by enhancing their animals’ genetic diversity [1]. Notably, horses exhibit rapid metabolic, anatomical, and physiological adaptations in response to extreme climates, particularly observed in populations inhabiting such regions [104]. Those overseeing domestic animal conservation play a crucial role in catalyzing regulatory changes that facilitate adaptive processes. Through strategic breeding, they can identify genes associated with enhanced adaptability to extreme climates without compromising performance. This proactive approach is vital in mitigating the challenges posed by climate change, ensuring future genetic diversity and resilience. Such insights are transferable across domestic animal populations globally, aiding breeders and owners in fostering resilient animals. This approach not only enhances profitability but also ensures the long-term sustainability of endangered animal populations, enabling them to thrive in the face of future climate extremes.

## 5. Conclusions

The proposed model can serve as a valuable tool in guiding strategies for the conservation and genetic improvement of horses in response to climate change. By encouraging breeders to pair genetically diverse and unrelated individuals, this approach fosters adaptability to extreme climatic conditions while maintaining performance standards. Additionally, ensuring increased levels of genetic diversity may eventually lead to enhanced adaptability; hence, those responsible for domestic population conservation may account for rather diverse genetic resources to promote regulatory changes necessary for breed adaptation. This model provides a framework for owners of endangered domestic populations to plan breeding strategies that prioritize resilience and adaptation to extreme climates. Ultimately, this proactive approach aims to enhance profitability while ensuring the long-term sustainability of these populations.

## Figures and Tables

**Figure 1 animals-14-00659-f001:**
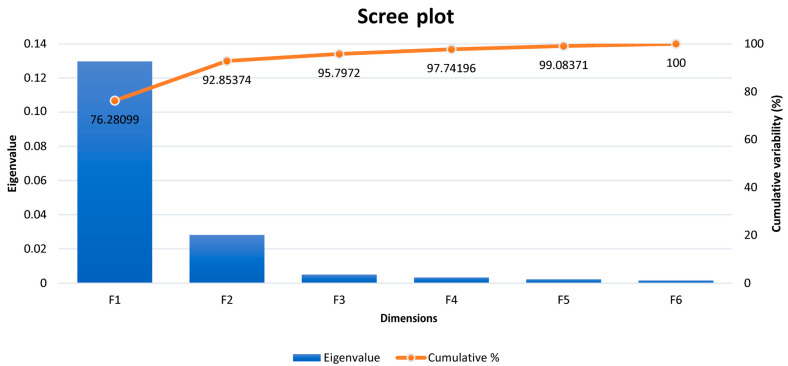
Eigenvalues and explained variability across dimensions identified by regularized canonical correlation analysis.

**Figure 2 animals-14-00659-f002:**
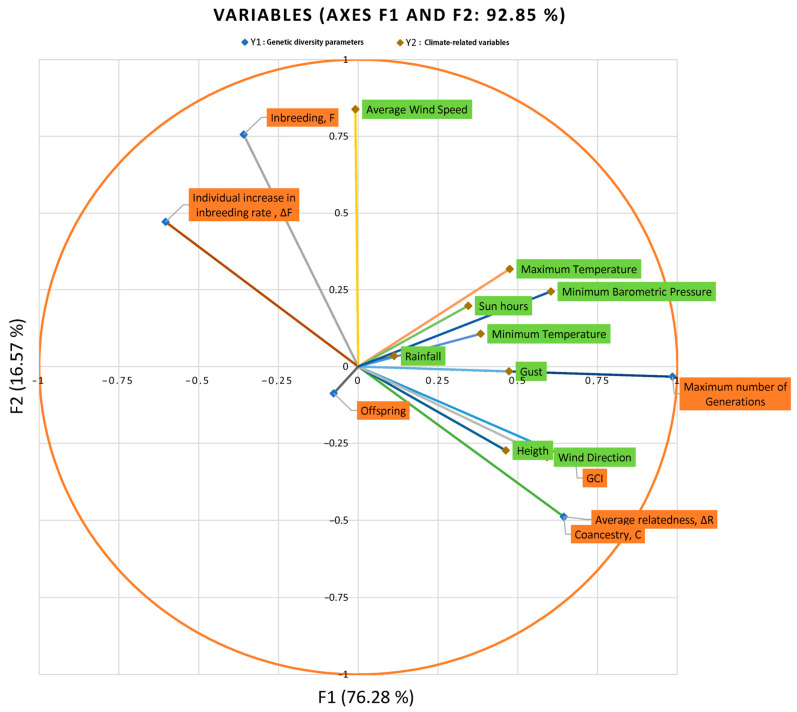
Canonical correlation component loadings.

**Figure 3 animals-14-00659-f003:**
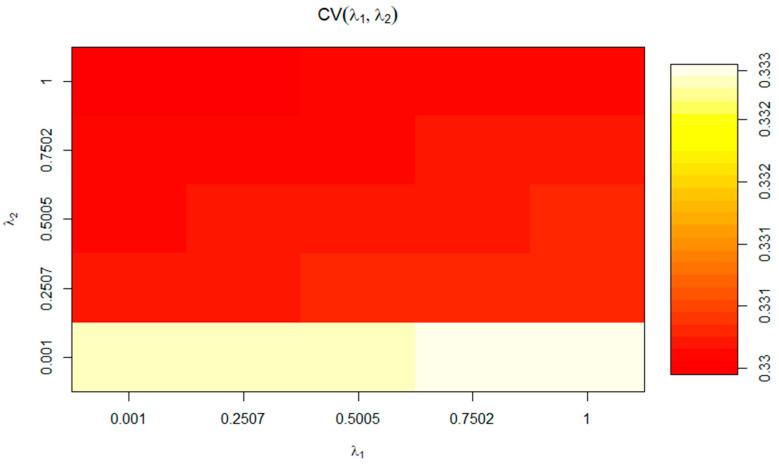
Optimal cross-validation scores for the values of the parameter of regularization (λ_1_ and λ_2_).

**Figure 4 animals-14-00659-f004:**
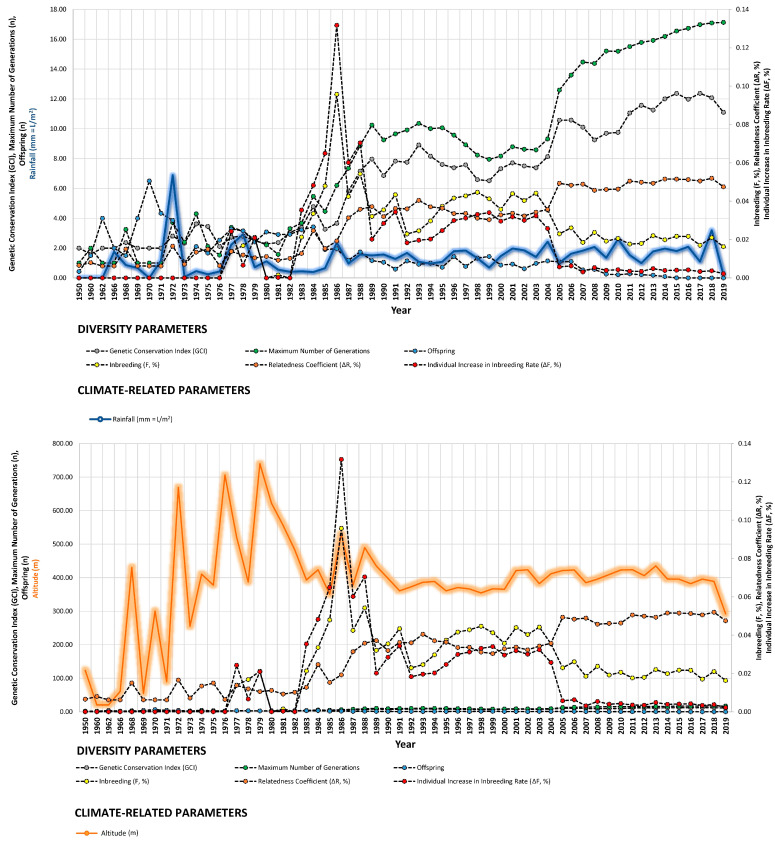
Trends of diversity parameters in relationship to rainfall (mm = L/m^2^), altitude (m), and minimum temperature (°C) from 1950 to 2019.

**Figure 5 animals-14-00659-f005:**
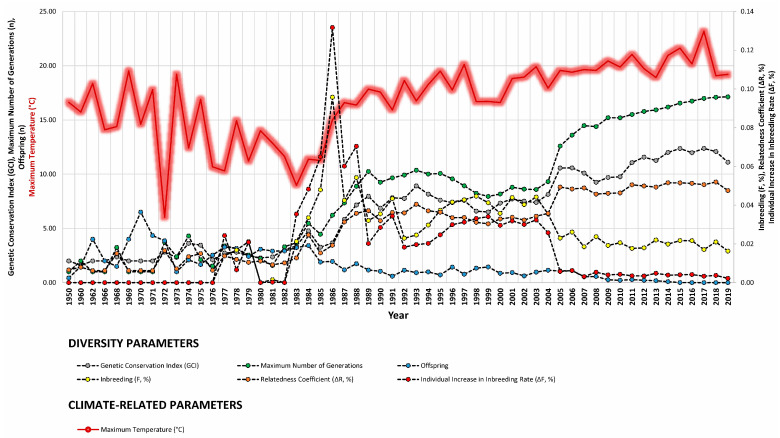
Trends of diversity parameters in relationship to maximum temperature (°C), wind direction (tenths of degree), and average wind speed (m/s) from 1950 to 2019.

**Figure 6 animals-14-00659-f006:**
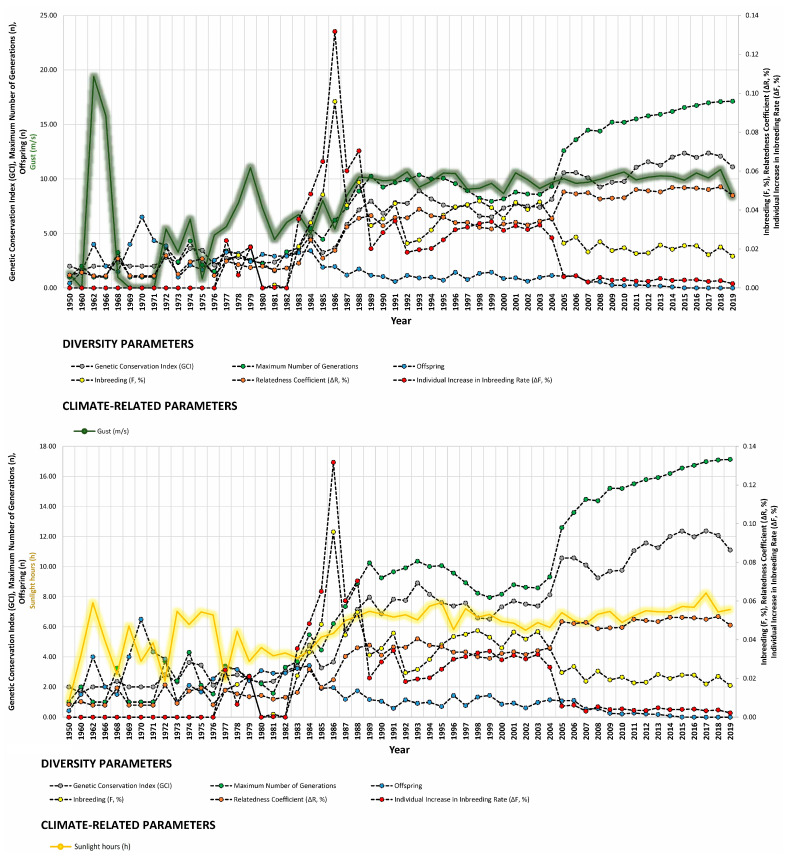
Trends of diversity parameters in relationship to gust speed (m/s), sunlight hours (h), and minimum barometric pressure (hPa) from 1950 to 2019.

**Table 1 animals-14-00659-t001:** Descriptive statistics for genetic diversity and climate-change-related parameters.

Cluster	Parameters	Mean	Mode	Std. Deviation	Variance	Minimum	Maximum	Percentile 25	Median/Percentile 50	Percentile 75
Genetic Diversity	Relatedness Coefficient (ΔR, %)	0.04	0.01	0.03	0.00	0.00	0.13	0.01	0.04	0.06
	Inbreeding (F, %)	0.03	0.00	0.05	0.00	0.00	0.50	0.00	0.00	0.06
	Genetic Conservation Index (GCI)	9.10	2.00	5.92	35.07	1.00	27.00	3.11	8.81	14.55
	Coancestry (C, %)	0.02	0.00	0.01	0.00	0.00	0.07	0.00	0.02	0.03
	Non-Random Mating Degree (α)	0.01	0.06	0.06	0.00	−0.05	0.49	−0.03	−0.01	0.03
	Number of Maximum Generations	12.14	16.00	5.75	33.05	0.00	21.00	11.00	15.00	16.00
	Number of Complete Generations	2.96	1.00	1.78	3.18	0.00	8.00	1.00	3.00	4.00
	Number of Equivalent Generations	6.16	2.00	3.04	9.25	0.00	11.54	3.89	6.76	8.79
	Individual Increase in Inbreeding Rate (ΔF, %)	0.02	0.00	0.04	0.00	0.00	0.47	0.00	0.00	0.01
	Offspring	0.73	0.00	3.76	14.10	0.00	219.00	0.00	0.00	0.00
Climate Change	Height (m)	399.77	172.71	341.50	116,624.62	0.00	1703.00	172.71	229.00	721.75
	Rainfall (mm)	1.64	0.00	4.60	21.12	0.00	57.80	0.00	0.00	0.56
	Minimum Temperature (°C)	8.13	0.00	5.42	29.33	−12.00	25.00	4.41	8.18	11.70
	Maximum Temperature (°C)	18.92	0.00	7.19	51.73	−3.60	44.55	14.66	18.62	23.34
	Wind Direction (Tenths of Degree)	18.29	0.00	13.18	173.69	0.00	99.00	5.23	19.33	26.67
	Average Wind Speed (m/s)	7.22	0.00	10.72	114.91	0.00	99.00	2.12	3.30	5.70
	Gust Speed (m/s)	9.78	0.00	4.04	16.33	0.00	33.13	7.50	9.50	11.90
	Sunlight Hours (h)	6.68	0.00	4.40	19.37	0.00	14.60	2.50	7.70	10.30
	Minimum Barometric Pressure (HPa)	916.20	0.00	228.16	52,055.31	0.00	1036.10	925.37	985.70	1003.80

**Table 2 animals-14-00659-t002:** Pearson’s correlations between genetic diversity parameter pairs. Color scale ranges from green (maximum positive value) to red (maximum negative value).

Variables	Inbreeding (F, %)	Relatedness Coefficient (ΔR, %)	Genetic Conservation Index (GCI)	Maximum number of Generations	Individual Increase in Inbreeding Rate (ΔF, %)	Offspring
Inbreeding (F, %)	1	−0.281	−0.301	−0.308	0.774	−0.003
Relatedness Coefficient (ΔR, %)		1	0.778	0.724	−0.406	−0.023
Genetic Conservation Index (GCI)			1	0.726	−0.414	−0.024
Number of Maximum Generations				1	−0.506	−0.036
Individual Increase in Inbreeding Rate (ΔF, %)					1	0.003
Offspring						1

**Table 3 animals-14-00659-t003:** Pearson’s correlations between climate-related parameter pairs. Color scale ranges from green (maximum positive value) to red (maximum negative value).

Variables	Altitude	Rainfall	Minimum Temperature	Maximum Temperature	Wind Direction	Average Wind Speed	Gust Speed	Sunlight Hours	Minimum Barometric Pressure
Altitude	1	−0.006	−0.345	−0.148	0.037	−0.011	0.070	0.106	0.041
Rainfall		1	0.018	−0.174	0.009	0.038	0.265	−0.338	0.035
Minimum Temperature			1	0.786	0.098	0.059	0.201	0.158	0.193
Maximum Temperature				1	0.069	0.081	0.103	0.503	0.318
Wind Direction					1	−0.438	0.208	0.182	0.237
Average Wind Speed						1	0.108	−0.114	0.030
Gust Speed							1	0.047	0.295
Sunlight hours								1	0.366
Minimum Barometric Pressure									1

**Table 4 animals-14-00659-t004:** Pearson’s correlations between genetic diversity and climate-related parameter pairs. Color scale ranges from green (maximum positive value) to red (maximum negative value).

Variables	Altitude	Rainfall	Minimum Temperature	Maximum Temperature	Wind Direction	Average Wind Speed	Gust Speed	Sunlight hours	Minimum Barometric Pressure
Altitude	1	−0.006	−0.345	−0.148	0.037	−0.011	0.070	0.106	0.041
Rainfall		1	0.018	−0.174	0.009	0.038	0.265	−0.338	0.035
Minimum Temperature			1	0.786	0.098	0.059	0.201	0.158	0.193
Maximum Temperature				1	0.069	0.081	0.103	0.503	0.318
Wind Direction					1	−0.438	0.208	0.182	0.237
Average Wind Speed						1	0.108	−0.114	0.030
Gust Speed							1	0.047	0.295
Sunlight hours								1	0.366
Minimum Barometric Pressure									1

**Table 5 animals-14-00659-t005:** Multivariate Tests of Significance.

Test Name	Value	Approx. F	Hypothesis df	Error df	Significance of F
Pillai’s trace criterion	0.15433	31.71974	54	64,878	0.001
Hotelling–Lawley trace criterion	0.17004	34.02836	54	64,838	0.001
Wilks’ lambda	0.85054	32.92284	54	55,114.8	0.001
Roy’s greatest root	0.11482				

F: Fisher–Snedecor; df: degrees of freedom.

**Table 6 animals-14-00659-t006:** Canonical correlations and Redundancy coefficients.

Variates	F1	F2	F3	F4	F5	F6	
Canonical correlations	0.339	0.166	0.071	0.057	0.048	0.039	
Squared Canonical Correlations	0.115	0.027	0.005	0.003	0.002	0.002	
Variates	F1	F2	F3	F4	F5	F6	Sum
Redundancy coefficients (Y1)	0.0459	0.0038	0.0008	0.0001	0.0002	0.0000	0.0510
Variates	F1	F2	F3	F4	F5	F6	Sum
Redundancy coefficients (Y2)	0.0214	0.0023	0.0005	0.0003	0.0001	0.0000	0.0247

**Table 7 animals-14-00659-t007:** Wilks’ Lambda test outputs for dimension reduction analysis.

Roots	Variates	Wilks’ Lambda	F	Hypothesis df	Error df	Sig. of F (Pr > F)
1 to 6	F1	0.85054	32.92284	54	55,114.8	0.001
2 to 6	F2	0.96087	10.83779	40	47,118.13	0.001
3 to 6	F3	0.98794	4.69107	28	38,977.43	0.001
4 to 6	F4	0.99289	4.2922	18	30,578.61	0.001
5 to 6	F5	0.99617	4.15126	10	21,624	0.001
6 to 6	F6	0.99844	4.21186	4	10,813	0.002

**Table 8 animals-14-00659-t008:** Univariate F-tests with (9;10,813) degrees of freedom (df).

Variable	Multivariate Generalization of R^2^	Adjusted R^2^	Hypothesis Mean Square	Error Mean Square	F	Significance of F (Pr < F)
Inbreeding (F, %)	0.02678	0.02597	0.08702	0.00263	33.05582	0.001
Relatedness Coefficient (ΔR, %)	0.05209	0.05130	0.04955	0.00075	66.02326	0.001
Genetic Conservation Index (GCI)	0.06655	0.06577	2806.35282	32.76502	85.65087	0.001
Maximum number of Generations	0.11066	0.10992	4399.28812	29.42751	149.49577	0.001
Individual Increase in Inbreeding Rate (ΔF, %)	0.04837	0.04758	0.10866	0.00178	61.07202	0.001
Offspring	0.00309	0.00226	1953.65319	524.39592	3.72553	0.001

## Data Availability

Data will be made available from the corresponding author upon reasonable request.

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
