# Peer review of "Modeling Climate Change Effects on Genetic Diversity of an Endangered Horse Breed Using Canonical Correlations"

_animals, 2024, doi:10.3390/ani14050659_

Round 1

Reviewer 1 Report

Comments and Suggestions for Authors

Dear Authors,

Your work is interesting and has new ideas, congratulations.

But it is long overall and needs to be reduced, it seems to come straight from an academic thesis for publication (it is important to publish the thesis studies, but they need to be changed before publishing).

The introduction is too long and needs to be shortened, the last paragraph should not be there, it should end the introduction, with a short sentence saying what your study is, like "The objective of the study was to evaluate the correlation and changes in genetic diversity and the influence of climate change, between the years 1955 to 2019." just an example.

In the material and methods, talk about all the details on each subject (good in a thesis) shouldn't even need to put it in an article, for example: Explain in detail what Person correlation is, it's not necessary, but there is more that you need to remove, if someone found it interesting, go see your Reference, and that's what references exist for.

Results, should not tell readers to go and see the tables as they do:

- Tables 2, 3 and 4 present the simple Pearson’s product moment correlations among 456 all variables specified.

- Table 6 and Supplementary Figure 1 report canonical correlations.

- Table 7 presents the six roots of RCCA. Additionally, Table 7 presents the results for...

They must put the minimum of what is in the table, and at the end (Table)

Discussion, point 4.1... should be reduced a lot to and I'm wondering if it should be here or in Material and methods, the same for 4.2. Now they put results in the discussion and they should be in the results, it cannot and should not include our results in the discussion it must be (Table) and the readers will see in results, they can and should include results of the studies (references) that they compare/indicate.

Too much discussion reports points of interest, but be careful, there are two moments that talk about one point and then repeat a little of the same at a later point, review.

Conclusion is also extensive and what you concluded must be written, the first paragraph is not a discussion, it can be in the abstract, summary, introduction and even discussion, but not in the conclusion. Nor parts ofpart of the second paragrade. They must say, based on what they studied, that the proposed model can help with strategies for the conservation and genetic improvement of horses in the face of climate change. 

In general, the study has information of interest for publication, but it is difficult to read (around 10 thousand words, the suggestion is 5 thousand a little more, but twice as much...), I believe that after reducing it, its interpretation improves, good job.

The Reviewer.

Author Response

Reviewer 1

Your work is interesting and has new ideas, congratulations.

But it is long overall and needs to be reduced, it seems to come straight from an academic thesis for publication (it is important to publish the thesis studies, but they need to be changed before publishing).

Response: We thank the reviewer for his/her kind comments, and we followed his/her suggestion to reduce the manuscript.

The introduction is too long and needs to be shortened, the last paragraph should not be there, it should end the introduction, with a short sentence saying what your study is, like "The objective of the study was to evaluate the correlation and changes in genetic diversity and the influence of climate change, between the years 1955 to 2019." just an example.

Response: Reviewer suggetsion was followed. Last paragraph was simplified and introduction was reducen from 1300 words to 985.

In the material and methods, talk about all the details on each subject (good in a thesis) shouldn't even need to put it in an article, for example: Explain in detail what Person correlation is, it's not necessary, but there is more that you need to remove, if someone found it interesting, go see your Reference, and that's what references exist for.

Response: We remove all the exceedign details as suggested by the reviewer. We agree with reviewers but most of the times reviewers ask fo rus to include all the information.

Results, should not tell readers to go and see the tables as they do:

- Tables 2, 3 and 4 present the simple Pearson’s product moment correlations among 456 all variables specified.

- Table 6 and Supplementary Figure 1 report canonical correlations.

- Table 7 presents the six roots of RCCA. Additionally, Table 7 presents the results for...

They must put the minimum of what is in the table, and at the end (Table):

Response: We do not agree wth the reviewers, as this presentation is clearer and is used in most of the articles. Anyway, we followed the reviewer suggestion.

Discussion, point 4.1... should be reduced a lot to and I'm wondering if it should be here or in Material and methods, the same for 4.2. Now they put results in the discussion and they should be in the results, it cannot and should not include our results in the discussion it must be (Table) and the readers will see in results, they can and should include results of the studies (references) that they compare/indicate. Too much discussion reports points of interest, but be careful, there are two moments that talk about one point and then repeat a little of the same at a later point, review.

Response: We followed the reviewer suggestions. Section 4.1. was reduced from 405 word to 257. Section 4.2. was reduced from 494 to 326. Section 4.3. was reduced from 1188 to 838. Section 4.4. was redcued from 259 to 156. Section 4.5. reduced from 1289 to 850.

Conclusion is also extensive and what you concluded must be written, the first paragraph is not a discussion, it can be in the abstract, summary, introduction and even discussion, but not in the conclusión.

Response: We followed the reviewer suggestion.

Nor parts of part of the second paragrade. They must say, based on what they studied, that the proposed model can help with strategies for the conservation and genetic improvement of horses in the face of climate change.

In general, the study has information of interest for publication, but it is difficult to read (around 10 thousand words, the suggestion is 5 thousand a little more, but twice as much...), I believe that after reducing it, its interpretation improves, good job:

Response: We thank the reviewer for his/her suggestions and considerably reduced the extension of the manuscript.

The Reviewer.

Reviewer 2 Report

Comments and Suggestions for Authors

Manuscript ID: animals-2842701
Title: A Canonical Correlation Tool to Model Climate Change Shaping Effects on the Genetic Diversity of an Endangered Horse Breed from 1950 to 2019

Review

The authors evaluated the impact of climate change on the genetic diversity of an endangered horse breed adapted to diverse Spanish climates. Using regularized canonical correlation analysis (RCCA), they traced the parallel evolution of the breed's genetic diversity and climate events from 1950 to 2019. Their findings were employed to develop predictive models for managing genetic diversity in response to future climate changes. Therefore, methodology can serve as a potential model for studying climate change effects on domestic animals globally.

I really enjoy reading this well composed paper, though it is too long in my opinion even as online publication. I wonder, if authors could think about forming an Appendix, where at least part of the statistical methods might be placed, along with the Supplementary files, leaving no Supplements, but being not in the main text? This would aid to better reading experience.

Tables 5–8 also could go to the Appendix as Tables S2–S5?

I also would think about the term “gusts” used as a factor. Do these gusts have a dimension? How this factor is related to the wind speed? "Wind Speed" is a general term that encompasses the overall speed of the wind. However, if the study or analysis specifically focuses on gusts, it may be mentioned separately as "Maximum Wind Gusts" or "Gust Speed." Wind gusts represent short bursts of higher wind speeds within a period of otherwise lower wind speeds and can be relevant in assessing the potential impact of strong, sudden winds on ecological systems.

By the way, in the Keywords there is “wind speed”.

Paper is definitely worth publishing, though there are other small comments that need to be answered in revision before acceptance:

1.       Title is definitely too long., please make it shorter

2.       Line 15: corresponding author data missing

3.       Line 22: should read “1950–2019”

4.       Line 203: perhaps, mistype?

5.       “XLSTAT 2014 (Pearson Edition) (Addinsoft, Paris, France)” repeated several times, try refer by reference number where possible, or use “XLSTAT 2014” after the first mention

6.       Line 449: should BE discarded?

7.       Table 1: add dimensions where possible. In particular, what about “gust”? Somehow must be measurable.

8.       Line 459 and below: “(0.000 < rXY < -0.300)” should read “(0.000 < rXY < –0.300)”?

9.       Table 2: add explanation of colors used into caption. Negative sign is “–“, not “-“

10.   Tables 3 and 4: as above

11.   Figure 1: I expect this is a “scree plot”?

12.   Figure 4–6, part of the text not readable. Try to increase font or use landscape format for a bigger size

13.   Back matter: check Template and add the role of funders to “Conflicts of interest”

14.   References: use en dash for the page range; add DOI where possible

Author Response

Review

The authors evaluated the impact of climate change on the genetic diversity of an endangered horse breed adapted to diverse Spanish climates. Using regularized canonical correlation analysis (RCCA), they traced the parallel evolution of the breed's genetic diversity and climate events from 1950 to 2019. Their findings were employed to develop predictive models for managing genetic diversity in response to future climate changes. Therefore, methodology can serve as a potential model for studying climate change effects on domestic animals globally.

I really enjoy reading this well composed paper, though it is too long in my opinion even as online publication. I wonder, if authors could think about forming an Appendix, where at least part of the statistical methods might be placed, along with the Supplementary files, leaving no Supplements, but being not in the main text? This would aid to better reading experience.

Response: We thank the reviewer for his/her kind comments. We shorten the manuscript extensión following the suggestion by both reviewers.

Tables 5–8 also could go to the Appendix as Tables S2–S5?

Reponse: We understand the reviewer’s point but this tables comprhensively present the results of Canoncial Correlation Analysis, which is the main topic of the article, hence we feel it could be misleading.

I also would think about the term “gusts” used as a factor. Do these gusts have a dimension? How this factor is related to the wind speed? "Wind Speed" is a general term that encompasses the overall speed of the wind. However, if the study or analysis specifically focuses on gusts, it may be mentioned separately as "Maximum Wind Gusts" or "Gust Speed." Wind gusts represent short bursts of higher wind speeds within a period of otherwise lower wind speeds and can be relevant in assessing the potential impact of strong, sudden winds on ecological systems.

Response: We clarified this in the body text. The Word speed was added after gusts across the body text. Gusts speed was considered in the sense suggested by the reviewer. However, Average wind speed refers to the mean wind speed registered in the area. According to our results, gusts are rather related to wind direction in degrees, which may be indicative of patterns in the manner such wind currents are formed in the area.

By the way, in the Keywords there is “wind speed”.

Response: We added the word average to make the distinction.

The paper is definitely worth publishing, though there are other small comments that need to be answered in revision before acceptance.

Response: We thank the reviewer's kind comments.

  1. The title is definitely too long., please make it shorter:

Response: As recommended, the manuscript title has been shortened to "Modeling Climate Change Effects on Genetic Diversity of an Endangered Horse Breed using Canonical Correlation".

  1. Line 15: corresponding author data missing:

Response: The corresponding author information has been added to the manuscript.

  1. Line 22: should read “1950–2019”:

Response: Modified following the reviewer's instructions.

  1. Line 203: perhaps, mistype?

Response: Corrected.

  1. “XLSTAT 2014 (Pearson Edition) (Addinsoft, Paris, France)” repeated several times, try refer by reference number where possible, or use “XLSTAT 2014” after the first mention:

Response: Reviewer suggestion was followed.

  1. Line 449: should BE discarded?:

Response: Reviewer suggestion was followed.

  1. Table 1: add dimensions where possible. In particular, what about “gust”? Somehow must be measurable.

Response: Reviewer suggestion was followed.

  1. Line 459 and below: “(0.000 < rXY < -0.300)” should read “(0.000 < rXY< –0.300)”?

Response: Reviewer suggestion was followed.

  1. Table 2: add explanation of colors used into caption. Negative sign is “–“, not “-“:

Response: Reviewer suggestion was followed. Colour scale ranges from green (maximum positive value) to red (maximum negative value).

  1. Tables 3 and 4: as above:

Response: Reviewer suggestion was followed. Colour scale ranges from green (maximum positive value) to red (maximum negative value).

  1. Figure 1: I expect this is a “scree plot”?:

Response: Yes. We corrected it.

  1. Figure 4–6, part of the text not readable. Try to increase font or use landscape format for a bigger size:

Response: We followed the reviewer suggestion.

  1. Back matter: check Template and add the role of funders to “Conflicts of interest”

Response: The role of funders was included in the “Conflicts of interest”.

  1. References: use en dash for the page range; add DOI where posible

Response: We followed the reviewer suggestion. DOIs are not compulsory, and due to the nature of the paper normally difficutl to find.

Round 2

Reviewer 1 Report

Comments and Suggestions for Authors

Dear Authors,Thank you for your answers and the improvement in your study, congratulations.

As I mentioned previously, your study is interesting and brings good ideas for the improvement and conservation of horses.

Having enough results, I understand that this requires a long discussion... this is to say that, I observed your attempt to reduce the study, but it was reduced to just 80 lines (600 words), the size of the article being the responsibility of the magazine (Editor), I am not opposed to the fact that it remains extensive.

Since it could be reduced in some moments of the discussion, I understand that you want to leave it to your study, explaining what each point refers to, some things that do not need to be written.

Line 596 to 601: I still think it's material and methods (not that it's bad here, but it explains that you chose this or that result in Material and methods).

Line 602 to 603: The following sentence is one that might cut: "The information provided by complete and equivalent generations overlaps with the maximum number of traced generations, representing the number of generations between an individual and its furthest known ancestor, even if that generation is incomplete due to unknown ancestors [63]."; I leave it to your discretion and that of the Editor.

The conclusion has improved, I just question the phrase that appears: "identifying associated genes" at no point in your text is this mentioned, because it appears here...

Continued good research,

The Reviewer.

Author Response

Dear Authors, Thank you for your answers and the improvement in your study, congratulations.

As I mentioned previously, your study is interesting and brings good ideas for the improvement and conservation of horses.

Having enough results, I understand that this requires a long discussion... this is to say that, I observed your attempt to reduce the study, but it was reduced to just 80 lines (600 words), the size of the article being the responsibility of the magazine (Editor), I am not opposed to the fact that it remains extensive.

Since it could be reduced in some moments of the discussion, I understand that you want to leave it to your study, explaining what each point refers to, some things that do not need to be written.

Response: We thank the reviewer for his/her kind comments.

Line 596 to 601: I still think it's material and methods (not that it's bad here, but it explains that you chose this or that result in Material and methods).

Response: We understand the reviewer’s point. However, here we discussed the rationale behind multicollinearity problems, that is explaining why our particular multicollinearity problems may have occurred. For researchers familiar to the area it may be not necessary, we agree, but for other readers, it may help to understand this and maybe other papers. We clarified this in the body text.

Line 602 to 603: The following sentence is one that might cut: "The information provided by complete and equivalent generations overlaps with the maximum number of traced generations, representing the number of generations between an individual and its furthest known ancestor, even if that generation is incomplete due to unknown ancestors [63]."; I leave it to your discretion and that of the Editor.

Response: We decide to leave it here for the same reasons as above.

The conclusion has improved, I just question the phrase that appears: "identifying associated genes" at no point in your text is this mentioned, because it appears here...

Response: We agree with the reviewer and decided to change the information.

Continued good research,

The Reviewer.